# The nucleus serves as the pacemaker for the cell cycle

**Oshri Afanzar[1], Garrison K Buss[2], Tim Stearns[3,4], James E Ferrell Jr[1,5]\***

[1]Department of Chemical and Systems Biology, Stanford Medicine, Stanford, United States; [2]Department of Molecular and Cellular Physiology, Stanford Medicine, Stanford, United States; [3]Department of Biology, Stanford University, Stanford, United States; [4]Department of Genetics, Stanford Medicine, Stanford, United States; [5]Department of Biochemistry, Stanford Medicine, Stanford, United States

**Abstract** Mitosis is a dramatic process that affects all parts of the cell. It is driven by an oscillator whose various components are localized in the nucleus, centrosome, and cytoplasm. In principle, the cellular location with the fastest intrinsic rhythm should act as a pacemaker for the process. Here we traced the waves of tubulin polymerization and depolymerization that occur at mitotic entry and exit in *Xenopus* egg extracts back to their origins. We found that mitosis was commonly initiated at sperm-derived nuclei and their accompanying centrosomes. The cell cycle was ~20% faster at these initiation points than in the slowest regions of the extract. Nuclei produced from phage DNA, which did not possess centrosomes, also acted as trigger wave sources, but purified centrosomes in the absence of nuclei did not. We conclude that the nucleus accelerates mitotic entry and propose that it acts as a pacemaker for cell cycle.

**\*For correspondence:**
james.ferrell@stanford.edu

**Competing interests:** The authors declare that no competing interests exist.

## Introduction

Mitotic entry is driven by a circuit of proteins that regulates cyclin-dependent protein kinase-1 (Cdk1) and its opposing phosphatases. The circuit includes at least five interlinked positive and double-negative feedback loops and functions as a bistable toggle switch, which transitions from a stable interphase state with low Cdk1 activity and high PP2A-B55 activity, to a stable M-phase state with high Cdk1 activity and low PP2A-B55 activity (*Sha et al., 2003*; *Pomerening et al., 2003*; *Mochida et al., 2009*; *Mochida et al., 2016*; *Vinod et al., 2013*). In some contexts, like the *Xenopus laevis* embryonic cell cycle, the cell cycle operates as a relaxation oscillator (*Tsai et al., 2008*), with the bistable trigger firing, and the cell cycle repeating, with a precise (± a few percent) (*Anderson et al., 2017*; *Newport and Kirschner, 1982a*; *Newport and Kirschner, 1982b*) fixed period.

Systems with a bistable trigger have the potential to generate trigger waves, propagating fronts of activity that spread outward from a source without slowing down or diminishing in amplitude (*Tyson and Keener, 1988*; *Gelens et al., 2014*). Familiar examples of biological trigger waves include the action potential (*Hodgkin and Huxley, 1952*), calcium waves (*Cornell-Bell et al., 1990*; *Gilkey et al., 1978*; *Stricker, 1999*), and apoptotic waves (*Cheng and Ferrell, 2018*). As predicted in the 1990s (*Novak and Tyson, 1993*), mitosis spreads through the cytoplasm via constant velocity (~60 µm/min) trigger waves (*Chang and Ferrell, 2013*).

A trigger wave typically originates from wherever the oscillator has the fastest intrinsic rhythm. For example, heartbeats originate at the sinoatrial node, which has a typical frequency of 60–90 $min^{-1}$ in humans, and spread to and override the slower rhythms of the atrioventricular node (whose intrinsic frequency is 40–60 $min^{-1}$) and ventricles (30–40 $min^{-1}$) (*Pappano and Wier, 2019*).

At present, it is uncertain where mitotic trigger waves begin. Plausible candidates include the centrosome, where cyclin B1-Cdk1 and Cdc25C have been found to concentrate in late interphase

(*Jackman et al., 2003*; *Bonnet et al., 2008*); the nucleus, to which cyclin B1-Cdk1 and Cdc25C translocate just prior to prometaphase (*Hagting et al., 1999*; *Yang et al., 2001*; *Santos et al., 2012*); or the cytoplasm, since cyclin A2-Cdk1/2 may shift from the nucleus to the cytoplasm late in interphase (*De Boer et al., 2008*; *Pines and Hunter, 1991*; *Meraldi et al., 1999*; *Li et al., 2010*). However, studies with FRET probes microinjected into mammalian cells have suggested that mitosis might be initiated in all parts of the cell essentially simultaneously (*Gavet and Pines, 2010*). This may well be true for somatic cells, where distances are on the order of 10 µm, whereas in larger cells like frog eggs it is clear that mitotic events happen in different parts of the cell at different times (*Chang and Ferrell, 2013*; *Hara et al., 1980*).

Here we set out to determine which subcellular compartment serves as the pacemaker for mitotic oscillations, through video microscopy studies of cycling *Xenopus laevis* egg extracts in a 96-well plate format. We looked for waves of mitotic entry, traced the waves back to their sources, and tested the putative sources through reconstitution experiments. These experiments demonstrate that the nucleus acts as the pacemaker for the *Xenopus* embryonic cell cycle.

## Results

### Cell cycles in thin layers of extract

To characterize mitosis in egg cytoplasm, we placed small volumes (3 µl) of egg extract in wells of a 96-well plate under mineral oil (*Figure 1A*). Cell-cycle dynamics in the resulting thin (~200 µm) layers of extract were monitored by time-lapse video microscopy. With this approach, up to 16 samples could be documented in the same imaging session while maintaining an acquisition frequency that was adequate to trace mitotic dynamics. Initially we included small concentrations (<10 sperm/mm$^2$) of demembranated sperm in the extracts. The sperm provided centrosomes, which promote microtubule organization, and chromatin, which recruits membranes from the extract and forms functional nuclei (*Lemaitre et al., 1998*; *Forbes et al., 1983*).

We used two probes to monitor cell-cycle progression: mCherry fused to a nuclear localization sequence (NLS-mCherry) and SiR-tubulin. During interphase, the NLS-mCherry accumulates in the reconstituted nuclei, and it disperses when the nuclear envelope breaks down at the onset of mitosis. SiR-tubulin is a docetaxel derivative that increases in fluorescence when bound to microtubules (*Lukinavičius et al., 2014*), allowing visualization of the oscillation between interphase (high microtubules) and mitotic (low microtubules) arrays. Oscillations in microtubule fluorescence were temporally coupled with the cycles of NLS-mCherry accumulation and dispersion, as expected (*Figure 1B*, *Figure 1—video 1*).

After warming to room temperature, the extract began to self-organize into cell-like compartments (e.g. *Figure 1—video 1*, *Figure 1C*, top left image; SiR-tubulin fluorescence is shown in red) (see also *Wang et al., 2013*; *Cheng and Ferrell, 2019*). During the first interphase, this organization occurred only around sperm and/or centrosomes. The partially-organized extract then underwent mitosis, as detected by a decrease in the overall concentration of microtubules (*Figure 1C*). As the extract exited mitosis and entered the second interphase, the entire well self-organized into cell-like compartments, even in regions that were not near sperm and/or centrosomes (*Figure 1C*, second row). Nuclei became visible about midway through interphase 2 (*Figure 1C*, shown in green).

Once nuclei could be seen, they and their associated cell-like compartments generally divided during each mitosis and therefore exponentially increased in number (*Figure 1D*). The fold-increase in the number of nuclei per cycle was typically less than a full doubling—1.71 ± 0.01 fold, mean ± S. D., for the experiment shown here. Notably, only sperm-associated compartments, with their centrosomes, divided.

### Spatial dynamics of mitotic initiation

The first mitosis in the extract shown in *Figure 1* and *Figure 1—video 1* occurred in a nearly concerted manner (*Figure 1C*, top row), with mitosis spreading quickly and irregularly through the whole well. Similar results were reported for extracts imaged in Teflon tubes (*Chang and Ferrell, 2013*). This spread most likely represents a phase wave—a type of wave that arises from differences (in this case small differences) in intrinsic timing rather than spatial coupling, like the waves of flashing lights on a movie marquee (*Tyson and Keener, 1988*; *Winfree, 1974*). Mitotic waves in the

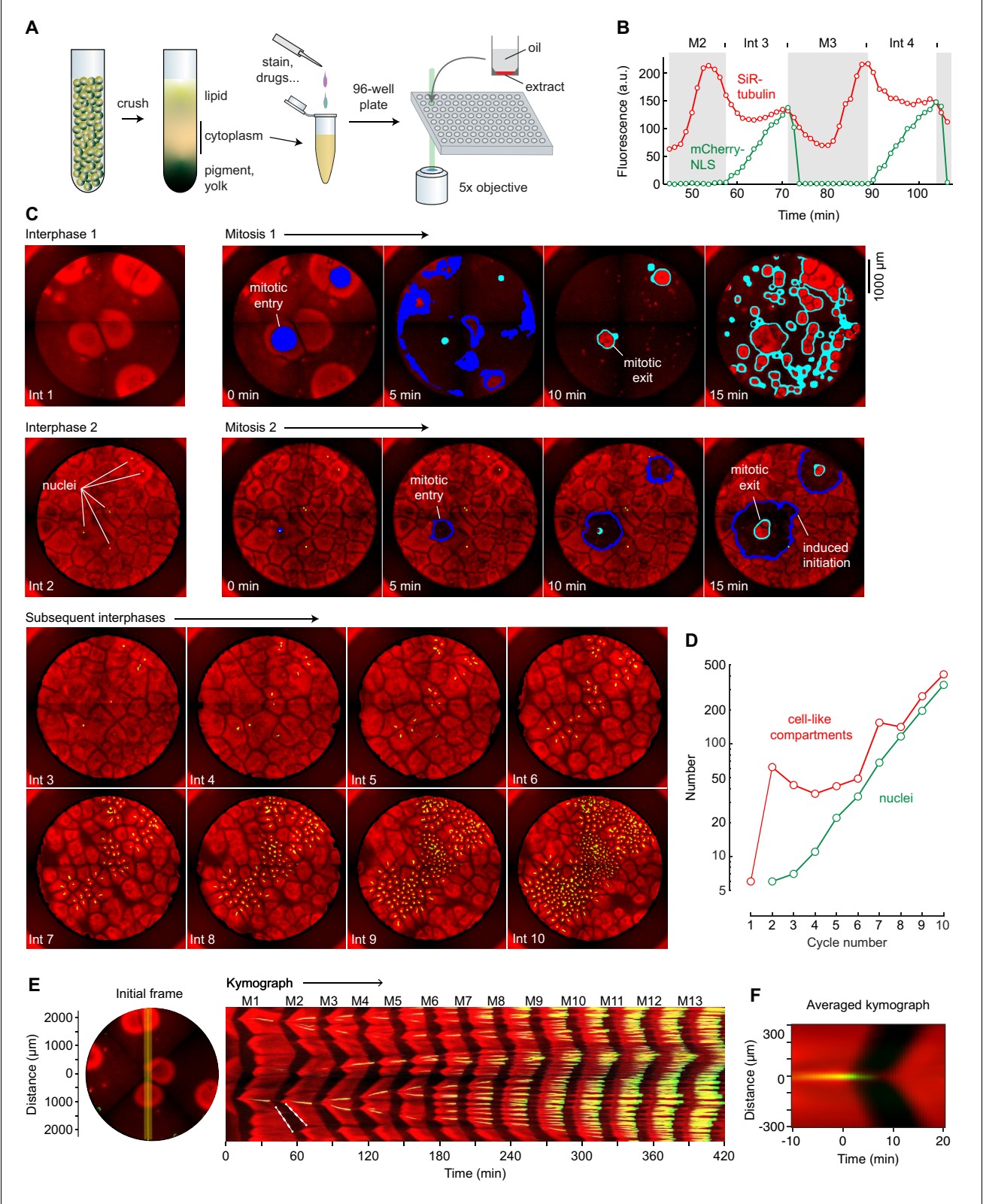

**Figure 1.** Cycling *Xenopus* egg extract resolved in two dimensions. (**A**) Experimental scheme. (**B–E**) Analysis of a typical cycling extract. (**B**) Comparison of NLS-mCherry (green) and SiR-tubulin (red) fluorescence through two cycles. Fluorescence was measured in a small region centered on where mitosis 2 first initiated. (**C**) Time course of the cycling extract. Snapshots of the first 10 interphases are shown together with montages of mitosis 1 and mitosis 2. Nuclei (NLS-mCherry) are shown in green and microtubules (SiR-tubulin) in red. The advancing fronts of mitosis are depicted in blue and the

*Figure 1 continued on next page*

*Figure 1 continued*

advancing fronts of interphase in cyan. (D) The number of nuclei (green) and cell-like compartments (red) as a function of cycle number. For cycle 1, we have included only the six large cell-like compartments. (E) A kymograph (right) from a cross section of an extract over 14 mitotic cycles. The dashed white line shows linear spread of microtubule depolymerization at the onset of mitosis, with a speed of 65 µm/min, and aster growth during the transition back to interphase, with a speed of 40 µm/min. The snapshot on the left depicts the region where the kymograph was calculated (yellow vertical line). Note that the data for panels B–E all came from the same experiment. (F) Radial kymograph from 33 mitoses from different experiments. The average speed of the front of microtubule depolymerization at mitotic entry was 54 ± 13 µm/min, and speed of the front of microtubule polymerization during mitotic exit was 32 ± 6 µm/min (mean ± S.D).

The online version of this article includes the following video and figure supplement(s) for figure 1:

**Figure supplement 1.** Heat maps showing the dynamics of mitosis for the experiment shown in *Figure 1*.

**Figure 1—video 1.** Trigger waves of tubulin depolymerization and polymerization in Xenopus extract supplemented with sperm.

https://elifesciences.org/articles/59989#fig1video1

**Figure 1—video 2.** Close up view of a dividing nuclear trigger wave source.

https://elifesciences.org/articles/59989#fig1video2

---

*Drosophila* embryo have also been attributed to phase waves of Cdk1 activity, and theory predicts a switch from phase waves to trigger waves as the cell cycle slows down (*Deneke et al., 2016*; *Vergassola et al., 2018*).

In contrast, during the second cycle (*Figure 1C*, second row) and subsequent cycles, mitosis generally initiated from well-defined sources and spread outward via regular circular waves of microtubule depolymerization. These waves expanded with constant speed and could be seen as linear fronts on kymographs (*Figure 1E*). For example, at the start of mitosis 2 in this experiment, waves of microtubule depolymerization originated in the vicinity of two nuclei and spread outward at a speed of 65 µm/min, similar to previously-reported mitotic wave speeds (*Chang and Ferrell, 2013*). This was followed about 10 min later by a slower (40 µm/min) wave of microtubule polymerization that initiated from a microtubule aster in the same location.

For 33 events from different experiments where mitotic waves clearly emerged from newly-formed nuclei, we produced radial kymographs; *Figure 1F* shows the average of the 33 kymographs. The mitotic wave velocity was 54 ± 13 µm/min (mean ± S.D.), similar to speeds reported previously (*Chang and Ferrell, 2013*). The microtubule polymerization waves at the beginning of interphase were slower, with a mean velocity of 32 ± 6 µm/min. This velocity is similar to that reported by Ishihara and colleagues for aster growth (22 ± 3 µm/min) (*Ishihara et al., 2016*). The waves of microtubule depolymerization and repolymerization are taken to be trigger waves, and the constant speed at which they expand is determined by the rate at which the biochemical reactions of the bistable mitotic trigger proceed and the rate at which the mitotic regulators diffuse locally (*Tyson and Keener, 1988*; *Gelens et al., 2014*; *Chang and Ferrell, 2013*; *Showalter and Tyson, 1987*).

As the nuclei and their associated centrosomes divided and separated, so did the trigger wave sources (*Figure 1—video 1*). This underscores the tight correlation between one or both of these structures and the trigger wave sources.

The qualitative behavior present here was commonly seen in other replicates, but not in all. Replicates that did not follow this general pattern typically either ceased cycling after a few cycles, possibly due to apoptosis, failed to form nuclei until later cycles, or showed trigger waves only during later cycles. Data from all experiments used for this study have been deposited in the Stanford Digital Repository and are available for examination.

## Trigger wave sources

In the experiment shown in *Figure 1*, *Figure 1—video 1*, and *Figure 1—video 2*, mitosis initiated primarily from where nuclei and their associated centrosomes were located. However, this was not always the case. For example, sometimes mitosis began at a location at the edge of the well, with no nearby nucleus present (*Figure 2—figure supplement 1*). This phenomenon has been noted previously in studies of mitotic trigger waves and apoptotic trigger waves in *Xenopus* extracts (*Cheng and Ferrell, 2018*; *Forbes et al., 1983*), and it is seen in mathematical models of trigger waves as well (*Nolet et al., 2020*). Thus, there are at least two types of sources: nuclear sources and edge sources. Both nuclear and edge sources generally produced well-defined, circular or semicircular, respectively, trigger wave fronts.

Mitosis also sometimes initiated from locations that were neither close to a nucleus nor to the edge of the well (denoted 'other' in *Figure 2B*). These 'other' sources were often seen in the first mitosis (before the extract was fully organized) and late in subsequent mitoses, and they usually did not generate well-defined circular waves. Examples of all three types of mitotic sources are shown in *Figure 2—video 1*.

## Automated identification and analysis of mitotic sources

To get a comprehensive view of where and when mitosis initiated, we used an automated approach to identify, map, and characterize mitotic sources from time-lapse video data from 25 experiments. Mitotic waves were taken to be fronts of microtubule depolymerization that expanded from one frame to the next. By this definition, for the experiment shown in *Figure 2A*, there were 11 mitotic waves by the 38 min time point. Note that initially (at 35 min) there had been numerous microtubule-depleted areas that superficially resembled mitotic regions but did not expand with time, and so these were not classified as mitotic. The centroid of a mitotic region in its first frame was generally taken to be the mitotic source. There were a few instances where the region was dumbbell-shaped; in these cases we considered mitosis to have originated nearly simultaneously from two distinct sources.

We then constructed heat maps to show when mitosis began at different locations in the well for each cell cycle. The darker the color, the earlier the time at which mitosis happened at that source. The heat map for the cycle shown in *Figure 2A* is shown in *Figure 2B*, and for comparison, the heat maps for 10 cycles from the experiment shown in *Figure 1* are shown as *Figure 1—figure supplement 1*.

We identified mitotic sources for each cycle and sorted them into three classes—nuclear sources (*Figure 2B*, orange), edge sources, (*Figure 2B*, red), and other sources (*Figure 2B*, blue). We also identified the last 15% of the pixels to enter mitosis, which often were regions that appeared not to have been entrained by a trigger wave (*Figure 2B*, green). We used these late pixels as a benchmark and expressed the time at which mitotic entry was advanced at the various trigger wave sources relative to these late regions.

## Mitotic sources are often associated with nuclei and/or centrosomes

*Figure 2—video 2* and *Figure 2C* show how the sources evolved through the first 9 cycles of the experiment used for *Figure 2A and B*. In the first cycle, most of the extract entered mitosis at about the same time. Numerous mitotic sources were identified, mostly in the 'other' (blue) category, and the sources tended to have entered mitosis only slightly earlier than the surrounding cytoplasm. By the third cycle, nuclei had appeared, and many of them acted as sources for trigger waves. These nuclear sources did not appear to generally be positioned where the 'other' sources had been in the initial cycle; they were new sources rather than stronger versions of the initial sources. By the ninth cycle, the strongest nuclear trigger wave sources entrained large areas, whereas most of 'other' sources had disappeared.

We repeated the automated procedure for 107 cycles from 25 different experiments. Overall, 717/3140 (23%) of the sources were close (<100 µm) to nuclei and their accompanying centrosomes. This proportion was ~5 times higher than would be expected by random chance (*Figure 2D*). To further test the robustness of this association, we calculated p-values by bootstrapping for each of the cycles individually. In 69/107 of the cycles, the number of trigger wave sources found to be close (<100 µm) to a nucleus was greater than would be expected by chance (p<0.05) (inset, *Figure 2D*), supporting the significance of the association between trigger wave sources and nuclei/centrosomes. As the cell cycles proceeded, the proportion of trigger waves that emanated from nuclei/centrosomes rose (as did the number of nuclei), and the proportion not associated with any obvious structure fell (*Figure 2E*). The 'best' trigger wave sources (those that were in the earliest 10% of sources to appear in a cycle, which were also the sources that controlled the largest areas of cytoplasm) were generally associated with nuclei (*Figure 2F*).

## The cell cycle is accelerated by ~20% in the vicinity of nuclear sources

By definition, mitotic sources are regions where the cell cycle is faster than it is in the surrounding cytoplasm. One way to estimate how much faster the cycles are is to look at early cycles where

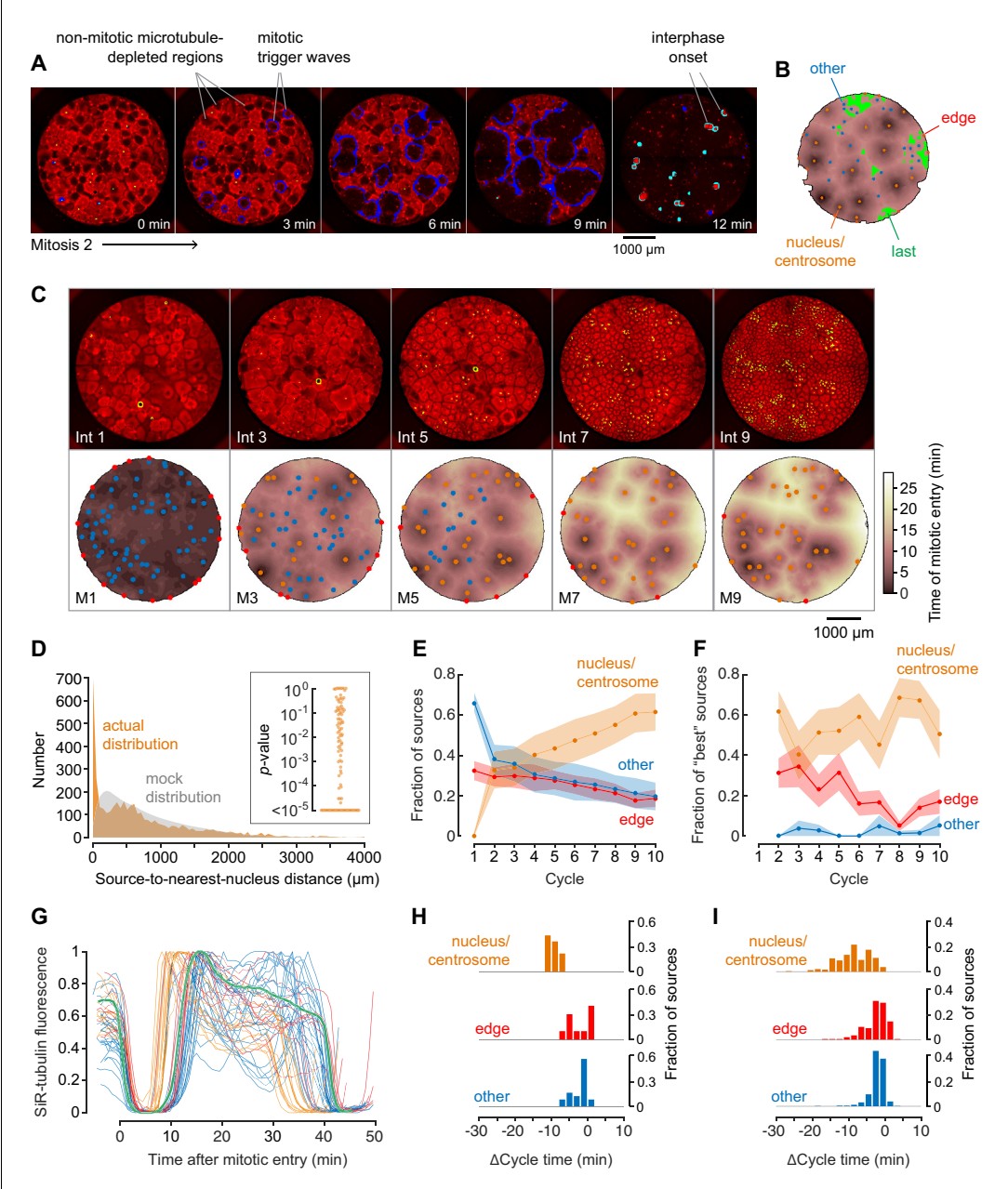

**Figure 2.** Demembranated sperm promotes mitotic initiation by shortening the cycle time. (A) Typical sperm-mediated mitotic trigger waves. Nuclei (NLS-mCherry) are shown in green and microtubules (SiR-tubulin) in red. The advancing fronts of mitosis are depicted in blue and the advancing fronts of interphase in cyan. (B) The trigger wave sources for the same cell cycle shown in (A), identified and characterized automatically. The time at which the various locations entered mitosis is depicted by a color scale, ranging from 46 min (dark) to 64 min (light). The latest 15% of the cytoplasm to enter mitosis is color-coded green. The trigger wave sources are plotted as filled circles of various colors: orange for sources associated with nuclei/centrosomes, red for sources at the edge of the well, and blue for others. (C) Montage showing changes in the pattern of mitotic sources from cycle 1 to 9. The coloring denotes the time at which each pixel entered mitosis relative to the earliest pixels in that cycle. (D) The distance of the trigger wave sources to the nearest nucleus (orange), compared to a mock distribution (gray). Data are from 107 cell cycles from 25 experiments and include 3140 sources. The inset shows the probabilities (p-values) of obtaining the observed number of mitotic sources (or more than the observed number) that are close (<100 μm) to nuclei for the 107 cell cycles analyzed individually. p-values were calculated by bootstrapping with $10^5$ randomized source positions for each individual cycle. (E) The fraction of the mitotic sources classified as associated with nuclei/centrosomes (orange), edges (red), or others (blue) as a function of cycle number. The data points are mean fractions from 13 experiments where the extracts cycled at least 10 times, and the shaded regions show the ± SEM. (F) The average fraction of 'best' sources (the earliest 10% of the sources) that were associated with nuclei/centrosomes (orange), edges (red), or others (blue) as a function of cycle number for the same experiments shown in panel E. (G) Time courses of microtubule fluorescence intensity, normalized to the maximum and minimum fluorescence in that cycle and time-aligned to make mitotic initiation occur at t = 0.

*Figure 2 continued on next page*

*Figure 2 continued*

The initiation of mitosis was taken as the time when the microtubule fluorescence first showed a significant decline. (H) Acceleration of the cell cycle, as measured by Δcycle time, at the different cell-cycle sources relative to the slowest 15% of the pixels for the experiment shown in (G). The Δcycle times were calculated as the time between the onset of the mitosis after nuclei appeared and the onset of the preceding mitosis, relative to the slowest 15% of the well. (I) Histograms of Δcycle times relative to the slowest 15% of the pixels for mitotic sources from 22 experiments. The total numbers of sources are 195 nucleus/centrosome sources, 182 edge sources, and 367 others.

The online version of this article includes the following video and figure supplement(s) for figure 2:

**Figure supplement 1.** Mitosis initiating from a source at the edge of a well.

**Figure 2—video 1.** Three classes of mitotic sources: others (left), nucleus-associated (center), and edge-associated (right).

https://elifesciences.org/articles/59989#fig2video1

**Figure 2—video 2.** Trigger waves of tubulin depolymerization and polymerization in a second Xenopus extract supplemented with sperm.

https://elifesciences.org/articles/59989#fig2video2

---

trigger waves are present in some of the well, but have not taken over the whole well, and compare the timing of the cell cycle at the trigger wave sources to that at the slower, unentrained regions. For example, in the kymograph shown in *Figure 1E*, at the onset of mitosis 2 there are two trigger wave sources, one near the top of the kymograph and one approximately 800 μm below the middle of the kymograph. The time from the start of M1 to the start of M2 at the two trigger wave sources was 35 and 39 min, respectively, whereas the slowest regions in the middle and at the bottom of the kymograph had cycles of 46 and 48 min, indicating that the cell cycle is on the order of ~10 min or ~20% faster at the nuclear trigger wave sources than in these slowest region.

The same type of analysis can be extended from the one-dimensional kymograph slice to the whole two-dimensional area of a well using the heat map representation. For the cycle shown in *Figure 2A and B*, the time courses of the changes in tubulin intensity at each of the mitotic sources and in the benchmark unentrained region are shown in *Figure 2G*. The latest regions of the cytoplasm for this experiment had an average cell-cycle duration of 40.5 ± 0.5 min (mean ± S.D, *Figure 2G*, green). Mitotic sources that were in proximity to nuclei entered mitosis 10.1 ± 1.4 min earlier, whereas most of the 'other' sources and the edge-associated sources were advanced by lesser amounts (3.4 ± 2.1 min and 3.5 ± 2.9 min, respectively) (*Figure 2G,H*). Thus, the nuclei and/or centrosomes appeared to be the most effective pacemakers, and they accelerated the cell cycle by 25 ± 3% in this experiment.

The same trend was seen in the aggregated data from 22 experiments (*Figure 2I*), with mitosis beginning from the nucleus-associated trigger wave sources 10 ± 4.4 min early, which is 20% quicker than the slowest regions of the well. Other classes of sources were accelerated by smaller amounts (3.5 ± 3.4 min for edge-associated sources and 2.9 ± 2.2 min for other sources, mean ± S.D.). These findings indicate that either the nucleus, the associated centrosomes, or the two structures together are particularly effective at accelerating the cell cycle and acting as a pacemaker structure. All told, 85% of the sources that accelerated mitosis by more than 10 min were nucleus-associated.

In summary, trigger waves that arise in the vicinity of nuclei and their associated centrosomes accelerate the cell cycle by approximately 10 min or 20%.

## Edge-associated sources are accelerated to a lesser extent

We next compared the behavior of these sperm-containing extracts to extracts with no added sperm chromatin, to determine where, in the absence of nuclei and centrosomes, mitosis was likely to originate. A typical experiment is shown in *Figure 3—video 1* and in montage form in *Figure 3A*. No cell-like compartments were formed during interphase 1, and the first mitosis occurred at all locations nearly simultaneously, although a few relatively shallow edge-associated sources and other sources were identified (*Figure 3A*). During the second interphase, cell-like compartments appeared, and two relatively strong edge sources emerged (at positions 12:00 and 1:00 on the heat map). These two edge sources grew to dominate the mitotic dynamics over the next several cycles; the self-organizing character of trigger waves (*Gelens et al., 2014*) is particularly apparent in these experiments where new sources are not being generated from nuclear division. A kymograph bisecting the stronger of the two sources showed a wave of microtubule depolymerization spreading at a constant speed of 60 μm/min (*Figure 3B*). For 28 different experiments the wave speed was 67 ± 17 μm/min (mean ± S.D.), similar to the speeds seen in the presence of sperm.

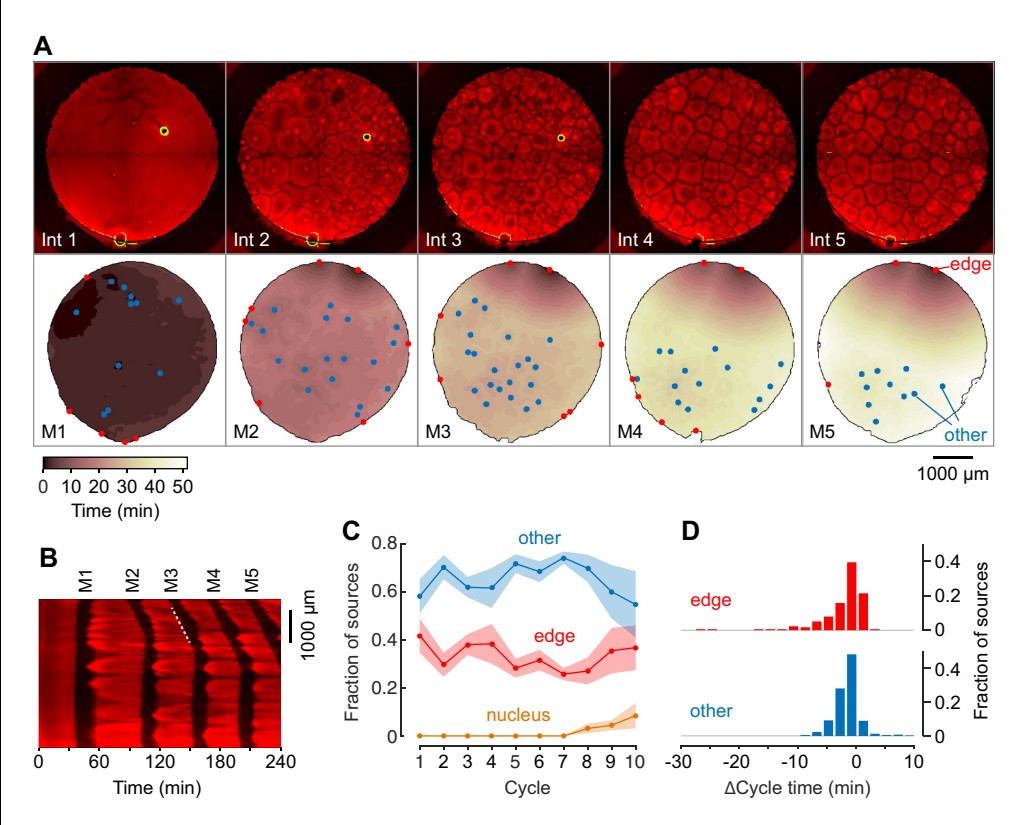

**Figure 3.** Mitotic waves in cycling extracts with no added sperm chromatin, DNA, or centrosomes. (**A**) Montage from a typical experiment showing the microtubules (top) and the heat map representation of the mitotic dynamics (bottom) for the first five cycles. Edge-associated sources are shown in red; other sources—that is, those not associated with edges or nuclei—are shown in blue (other). The greenish circle in the upper right-hand quadrant of the first three frames of the microtubule montage is a bubble, not a nucleus. The coloring of the heat map representation denotes the time at which each pixel entered mitosis relative to the earliest pixels in that cycle. See also *Figure 3—video 1*. (**B**) Kymograph based on *Figure 3—video 1*. The kymograph was calculated for a slice beginning at the edge source at 1:00 in (**A**) and transecting the well. The trigger wave speed at the entrance to M3 was 60 μm/min (dashed white line). (**C**) Mean fraction of the mitotic sources originating from edges, nuclei, and other locations as a function of cycle number. Data are from six experiments. (**D**) Acceleration of the cell cycle at edge sources and other sources relative to the slowest 15% of the well. Data are from 24 experiments and include 176 edge sources and 393 other sources.

The online version of this article includes the following video for figure 3:

**Figure 3—video 1.** Trigger waves of tubulin depolymerization and polymerization in a Xenopus extract supplemented with no sperm, centrosomes, or nuclei added.

https://elifesciences.org/articles/59989#fig3video1

From six independent experiments with at least 10 cycles, most of the sources were either associated with the edge of the well or with no obvious structure (*Figure 3C*). In some experiments, nuclei began to appear and replicate in the later cycles even though no sperm had been added (*Figure 3C*), presumably as a result of replication of small amounts of egg DNA present in the extracts. Most of the sources appeared to accelerate the cell cycle by very little; for the edge sources the average Δcycle time was −3.1 ± 5.3 min, with 10/569 (1.8%) of the edge sources, including the two dominating sources in *Figure 3A*, having Δcycle times of 10 min or more (*Figure 3D*). Overall the 'other' sources had an average Δcycle time of −2.4 ± 2.2 min, with only 1/569 (0.2%) having Δcycle times of 10 min or more (*Figure 3D*). Thus, extracts containing no nuclei or centrosomes cycled and generated mitotic trigger waves, but their mitotic sources were generally weaker than nucleus/centrosome-associated sources.

## Centrosomes promote cytoplasmic organization and division, but do not measurably accelerate mitosis

Sperm provide the extract with two plausible mitotic sources, centrosomes and nuclei. To determine whether centrosomes are sufficient to pace the cytoplasm, we added purified HeLa cell centrosomes to the extracts at concentrations of up to eight centrosomes/mm$^2$ and assessed their effect on mitotic dynamics. The added centrosomes produced foci of microtubules during interphase that promoted the rapid organization of cell-like compartments (*Figure 4A,B*). The extracts then cycled (shown in montage in *Figure 4C* and in *Figure 4—video 1*), with the centrosomes increasing in number during each cell cycle (*Figure 4C*) and the cell-like compartments dividing as they did in the presence of sperm (*Figure 4B*). Boiled centrosomes did not promote the organization of cell-like compartments, did not replicate, and did not promote the division of the cell-like compartments (*Figure 4—figure supplement 1*).

Nuclei sometimes eventually appeared in centrosome-supplemented extracts, even though no sperm chromatin had been added, and once they appeared, they replicated just as the centrosomes did (*Figure 4C,D*). The appearance of nuclei was inhibited by the DNA polymerase inhibitor aphidicolin in a dose-dependent fashion (*Figure 4E*), consistent with the hypothesis that they arose from the replication of small quantities of *Xenopus* egg DNA present in the egg extracts, or of HeLa cell DNA present in the purified centrosomes. The proportion of sources associated with nuclei increased with the cycle number (*Figure 4F*).

To determine whether centrosomes were acting as mitotic sources, we compared the distances between sources and the nearest centrosome to what would be expected by chance. The actual distribution of distances was similar to the mock distribution (*Figure 4G*), and only 3/18 individual cycles had more sources within 100 µm of a centrosome than would be expected by chance (*Figure 4G*, inset). Once nuclei appeared, some became trigger wave sources, and the number of sources close (<100 µm) to a nucleus was substantially higher than what would be expected by chance (*Figure 4H*; 66/149 of the individual cycles). As was the case with sperm-supplemented extracts, these nuclei accelerated the cell cycle more than centrosome-associated sources, edge-associated sources, or other sources did (*Figure 4I and J*).

Adding aphidicolin to centrosome-supplemented extracts did not alter the organization of the cytoplasm or the division of the cell-like compartments (*Figure 4K* and *Figure 4—video 2*), it but did suppress the appearance of nuclei and left the extract with mitotic sources that only modestly accelerated the cell cycle (*Figure 4K–M*).

Thus, although HeLa cell centrosomes promoted microtubule organization and allowed the cell-like compartments to divide, they acted as relatively weak mitotic sources; they did not accelerate the cell cycle to a greater extent than edge-associated sources or sources not associated with particular structures.

## Phage DNA forms nuclei that accelerate mitosis in the absence or presence of centrosomes, alike

To determine whether nuclei could substantially accelerate mitotic entry in the absence of centrosomes, we examined mitotic initiation in centrosome-less extracts that were supplemented with λ-bacteriophage DNA (*Figure 5A* and *Figure 5—video 1*). The phage DNA (nominally 5 µg/ml) formed nuclei, as previously reported (*Forbes et al., 1983*), and the nuclei replicated, but generally the cell-like compartments did not divide. This indicates that centrosomes are required for compartment division and that the production of nuclei from the phage DNA did not result in the *de novo* production of centrosomes.

Many, but not all, of the nuclei became mitotic sources, and the initially strong sources became more dominant (deeper wells on the heat map) with time (*Figure 5A–B*). Analysis of 44 cycles from 11 experiments showed that there were far more sources in proximity to nuclei than what would be expected by chance (*Figure 5C*). Out of the 44 cycles, 29 had more sources in close proximity (<100 µm) to nuclei than would be expected by chance (p<0.05).

We assessed the cell-cycle acceleration produced by the phage DNA by comparing the cycle times at the nuclear sources to the slowest 15% of pixels in the first cycle where nuclei were apparent. The distribution of Δcycle times for the three classes of sources is shown in *Figure 5D*. The average acceleration at nuclear sources was 6.6 ± 4 min (*Figure 5D*), about two-thirds of what was seen

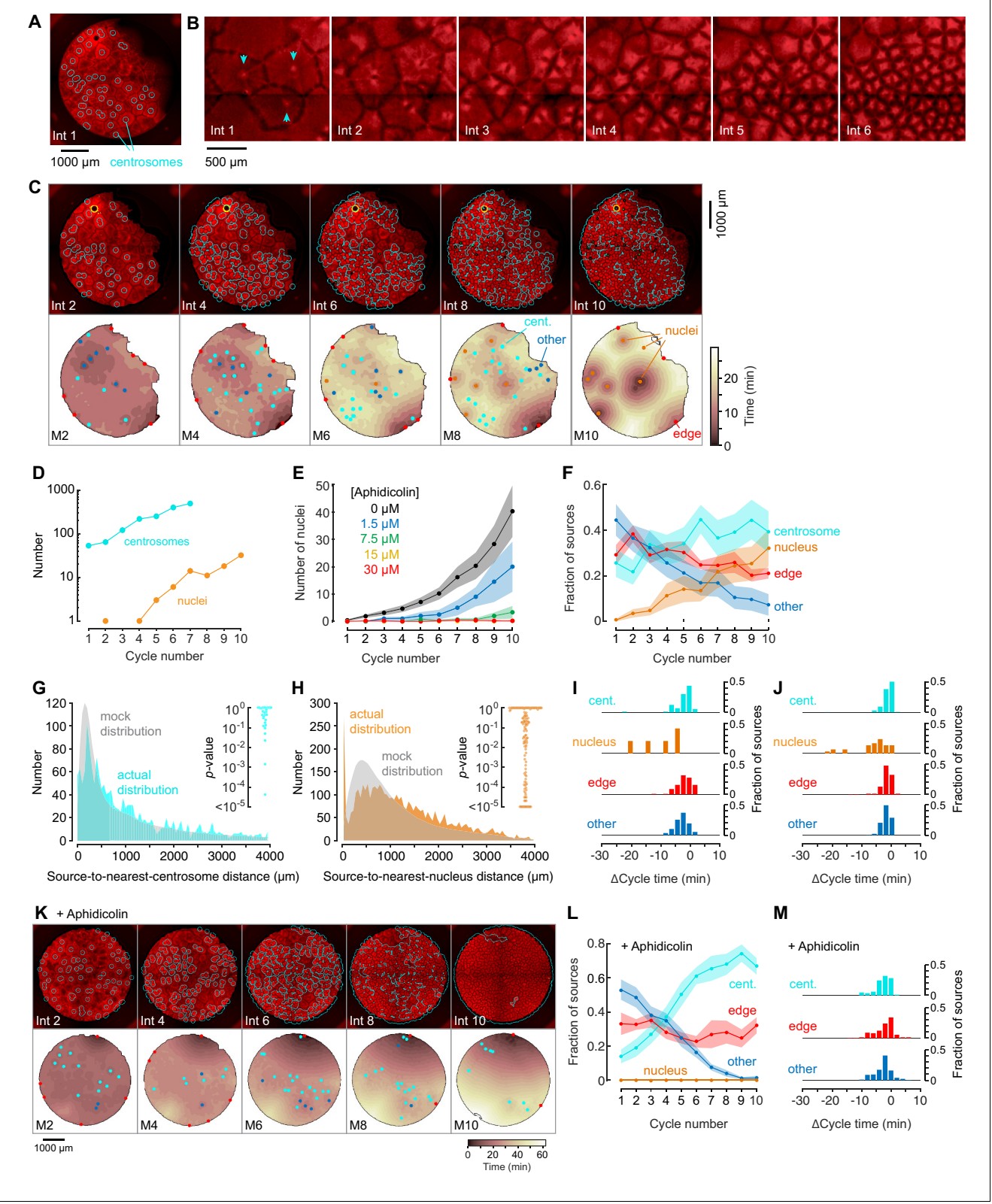

**Figure 4.** Centrosomes promote cytoplasmic organization and division of the cell-like compartments, but are not strong pacemaker sources. (**A**) Identification of centrosomes in an extract supplemented with purified HeLa cell centrosomes. (**B**) Montage showing that centrosomes duplicate and promote the division of cell-like compartments. SiR-Tubulin fluorescence is shown in red. (**C**) Montage showing SiR-tubulin fluorescence (red), NLS-mCherry fluorescence (green), and centrosome positions (cyan) (top), and heat maps of the mitotic sources (bottom), as a function of cycle number from

*Figure 4 continued on next page*

*Figure 4 continued*

a typical experiment. The coloring of the heat map denotes the time at which each pixel entered mitosis relative to the earliest pixels in that cycle. See also *Figure 4—video 1*. (D) The number of centrosomes and adventitiously-produced nuclei per cell cycle for the experiment shown in (C). (E) Suppression of nucleus formation by aphidicolin. (F) The fraction of the mitotic sources associated with nuclei/centrosomes (orange), edges (red), centrosomes (cyan) or others (blue) as a function of cycle number. The data points are mean fractions from 18 experiments where the extracts cycled at least 10 times, and the shaded regions show the ± SEM. (G, H) The distribution of distances from mitotic sources to the nearest centrosome (G, cyan), from 19 cycles, 11 experiments, and 494 sources, and from mitotic sources to the nearest nucleus (H, orange), from 149 cycles, 21 experiments, and 3993 sources. Only those cycles with fewer than 100 centrosomes or nuclei were included in G or H, respectively. The expected random distributions (mock distribution) of distances are shown in gray. The insets show probabilities (p-values) of obtaining the observed number of mitotic sources (or more than the observed number) that are close (<100 μm) to centrosomes (G) or nuclei for the individual analyzed cycles. p-values were calculated by bootstrapping with $10^5$ randomized source positions for each individual cycle. (I) Acceleration of the cell cycle (measured as Δcycle times) relative to the slowest 15% of the cytoplasm. These Δcycle times were calculated for the first cell cycle after the appearance of centrosomes. Data are from 17 experiments and include 74 centrosome-, five nucleus-, 103 edge-associated sources and 167 other sources. (J) Acceleration of the cell cycle (measured as Δcycle times) relative to the slowest 15% of the cytoplasm, calculated for the first cell cycle after the appearance of nuclei (for those experiments where nuclei appeared). Data are from 19 experiments and include 369 centrosome-, 28 nucleus-, 132 edge-associated and 123 other sources. (K) Montage showing SiR-tubulin (red), NLS-mCherry (green), and centrosomes (cyan) for a centrosome-supplemented extract treated with 15 μM aphidicolin. (L) The fraction of the mitotic sources associated with centrosomes (cyan), edges (red) or others (blue) as a function of cycle number for aphidicolin-treated (15 to 60 μM), centrosome-supplemented extracts. From 11 experiments where the extracts cycled at least 10 times. (M) The Δcycle times for the three classes of mitotic sources. Data are from 14 experiments and include 82 centrosome-, 121 edge-associated sources and 169 other sources.

The online version of this article includes the following video and figure supplement(s) for figure 4:

**Figure supplement 1.** Montage of an extract with added boiled centrosomes (A) or normal centrosomes (B).

**Figure 4—video 1.** Trigger waves of tubulin depolymerization and polymerization in a *Xenopus* extract supplemented with purified HeLa cell centrosomes.

https://elifesciences.org/articles/59989#fig4video1

**Figure 4—video 2.** Trigger waves of tubulin depolymerization and polymerization in a Xenopus extract supplemented with purified HeLa cell centrosomes and aphidicolin (15 μM).

https://elifesciences.org/articles/59989#fig4video2

with nuclear sources in sperm-supplemented extracts (*Figure 2I*), but higher than what was typically seen for edge sources, centrosome-associated sources, or other sources (*Figures 2–4*). The acceleration seen in the phage DNA experiments may be an underestimate; the nuclear sources initially present in the phage-supplemented extracts were so numerous that little if any of the extract appeared to not be entrained by a trigger wave. Alternatively, it is possible that nuclei formed around phage DNA have somewhat slower intrinsic rhythms than nuclei formed around sperm chromatin, or that nuclei lacking centrosomes have slower rhythms than nuclei that possess them.

To test this third possibility, we incubated extracts with purified HeLa cell centrosomes plus phage DNA and assessed mitotic dynamics. As shown in *Figure 5E* and *Figure 5—video 2*, the centrosomes facilitated the organization of the extract and allowed the cell-like compartments to divide, as expected. However, it appears that the presence of centrosomes had slowed down the acceleration of the cell cycle by the phage DNA to 4.1 ± 2.5 min (*Figure 5F*) as measured by the Δcycle times in 12 experiments. Thus, if there an additional effect of having centrosomes present as a staging platform for mitosis, it is too subtle for this type of analysis to detect.

## Discussion

Here we used *Xenopus* egg extracts and reconstitution to determine what structure acts as pacemaker for the mitotic oscillator. We found that nuclei that were formed from *Xenopus* sperm chromatin (*Figures 1–2*) or phage DNA (*Figure 5*) acted as robust sources of mitotic trigger waves. This could be attributed to an acceleration of the cell cycle by about 10 min. In the absence of nuclei, added centrosomes facilitated the organization of cell-like compartments and allowed the cell-like compartments to divide, but had little ability to act as trigger wave sources (*Figure 4*). These findings support the hypothesis that the nucleus normally serves as the cell-cycle pacemaker. It seems plausible that the ability of the nucleus to concentrate Cdc25C and cyclin B-Cdk1 (*Bonnet et al., 2008*; *Santos et al., 2012*; *Pines and Hunter, 1991*) could be the mechanistic basis for the pacemaking, although a direct experimental test of this hypothesis has not yet been carried out.

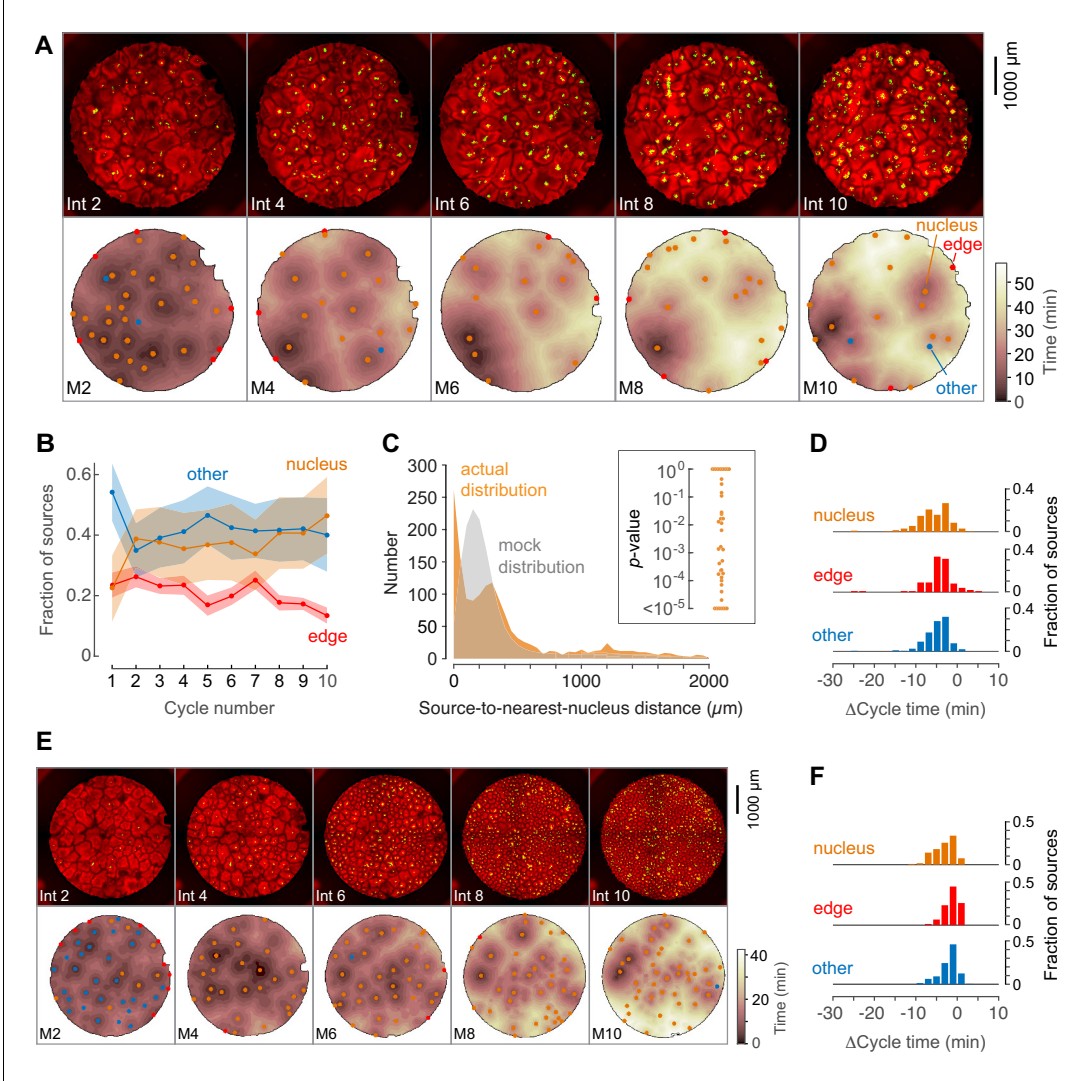

**Figure 5.** Phage DNA forms nuclei that can act as trigger wave sources. (**A**) Montage from a typical experiment showing the microtubules (top) and the heat map representation of the microtubule sources (bottom) for cycles 2, 4, 6, 8, and 10. Nucleus-associated sources are shown in orange; edge-associated sources are shown in red; and other sources—that is, those not associated with edges or nuclei—are shown in blue (other). The top panels show the extracts in interphase, with SiR-tubulin fluorescence in red and NLS-mCherry in green. The bottom panels are the heat maps of the mitotic sources. The coloring of the heat map denotes the time at which each pixel entered mitosis relative to the earliest pixels in that cycle. (**B**) Fraction of the mitotic sources associated with nuclei, edges, or no obvious structures as a function of cycle number. The data points are mean fractions from eight experiments where the extracts cycled at least 10 times, and the shaded regions show the means ± SEM. (**C**) The distance of the trigger wave sources to the nearest nucleus (orange), compared to a mock distribution (gray). Data are from 44 cell cycles from 11 experiments and include 419 sources. The inset shows probabilities (p-values) of obtaining the observed number of mitotic sources (or more than the observed number) that are close (<100 μm) to nuclei for the 44 cell cycles individually, calculated by bootstrapping with $10^5$ randomized source positions for each individual cycle. (**D**) The duration of the cell cycle at nucleus-associated sources, edge sources, and other sources relative to the slowest 15% of the well. Data are from 15 experiments and include 205 nucleus-associated sources, 102 edge sources and 201 other sources. (**E**) Montage from an experiment with added phage DNA plus centrosomes. (**F**) The duration of the cell cycle at nucleus-associated sources, edge sources, and other sources relative to the slowest 15% of the well, for extracts with added phage DNA plus centrosomes. Data are from 12 experiments and include 123 nucleus-associated sources, 79 edge sources and 209 other sources.

The online version of this article includes the following video(s) for figure 5:

**Figure 5—video 1.** Trigger waves of tubulin depolymerization and polymerization in a Xenopus extract supplemented with λ-bacteriophage DNA (5 μg/ml).

*Figure 5 continued on next page*

*Figure 5 continued*

https://elifesciences.org/articles/59989#fig5video1

**Figure 5—video 2.** Trigger waves of tubulin depolymerization and polymerization in a Xenopus extract supplemented with purified HeLa cell centrosomes and λ-bacteriophage DNA (5 µg/ml).

https://elifesciences.org/articles/59989#fig5video2

These studies underscore that diverse biological processes, for example heart beats and cell cycles, may occur on different time scales (~1 s vs. ~40 min) and involve different classes of proteins (ion channels vs. kinases, phosphatases, and ubiquitin ligases), but share an underlying unity in terms of their systems-level logic. In both cases the rhythms are generated by a relaxation oscillator circuit with a bistable trigger, and in both cases this allows the rhythm to spread in an orderly fashion from a spatially-distinct pacemaker locus via trigger waves.

That said, mechanisms other than trigger waves can be responsible for spatiotemporal activity waves, even in the regulation of mitosis. For example, in *Drosophila* embryos, mitosis sweeps through the embryo, but these waves have been attributed to a pre-existing spatial gradient (*Deneke et al., 2016*; *Vergassola et al., 2018*). Nevertheless, even in this system, the ingredients required for trigger waves—bistability plus a local coupling mechanism—are present, and it is predicted that the phase waves become trigger waves when the cycle slows (*Vergassola et al., 2018*).

What might be the advantages to a cell for having mitosis spread from a nuclear pacemaker as opposed to, say, from a focus on the cell periphery, or from some random position in the cytoplasm? It is possible that the mechanics of mitosis are more reliable when nuclear events such as chromatin condensation occur prior to cytoplasmic events such as the fragmentation of the endoplasmic reticulum and cortical events such as cell rounding. Alternatively, it is possible that the cell is simply taking advantage of the partitioning of mitotic regulators between the nucleus and the cytoplasm (e.g. Myt1 and PP2A are cytoplasmic whereas Wee1 and Gwl are nuclear [*Heald et al., 1993*; *Mueller et al., 1995*; *Liu et al., 1997*; *Wang et al., 2016*]) to add a spatial switch to the reactions of mitotic initiation (*Santos et al., 2012*; *Ferrell, 1998*). Just as reaction dynamics (e.g. bistability) can give rise to spatial patterns (e.g. trigger waves), spatial factors (e.g. regulated translocation) can contribute to reaction dynamics.

While this manuscript was in preparation, Nolet and colleagues published a paper that also argues that nuclei function as cell-cycle pacemakers in *Xenopus* extracts, based on a combination of theory and experiments in a one-dimensional extract system (*Nolet et al., 2020*). Our work agrees with, complements, and extends their findings by directly assessing the contribution of the centrosome to the pacemaker function.

# Materials and methods

**Key resources table**

| Reagent type (species) or resource | Designation | Source or reference | Identifiers | Additional information |
|---|---|---|---|---|
| Strain, strain background | *Xenopus laevis* | NASCO | LM00535MX | female |
| Cell line | HeLa centrin-GFP | *Tsou and Stearns, 2006* | | |
| Peptide, recombinant protein | GST-NLS-mCherry | *Chang and Ferrell, 2013* | | |
| Chemical compound, drug | SiR-tubulin | Spirochrome | CY-SC002 | |
| Chemical compound, drug | aphidicolin | Calbiochem | CAS 38966-21-1 | |
| Chemical compound, drug | λ-DNA | New England Biolabs | N3011S | |
| Software, algorithm | custom-made Matlab scripts | available at https://purl.stanford.edu/fm814ch0699 | | |
| Other | demembranated sperm | *Murray, 1991* | | |

## Extract preparation

Frog handling and egg extract preparation is as described in *Chang and Ferrell, 2018*, except that the extract was typically clarified 3 times for 8 min at 16,000 g to improve optical transparency. To

prevent the over-dilution of the extract, the addition of probes, cell components, or drugs, was typically done at a ratio of 1:100. SiR-Tubulin (Spirochrome, dissolved in DMSO) was used at 0.3 µM (0.3% final DMSO concentration). Aphidicolin (Calbiochem; dissolved in DMSO) was applied as indicated in text (0.16% final concentration of DMSO for every 10 µM of aphidicolin). GST-NLS-mCherry was purified and used as described in the Methods section of reference (*Cheng and Ferrell, 2018*). λ-DNA (New England Biolabs) was used at 0.5 µg per 100 µl of extract. Centrosomes were purified following *Mitchison and Kirschner, 1984*, and were typically applied at 30–1000 centrosomes per 100 µl of extract. Demembranated sperm was prepared as described (*Murray, 1991*), and was typically used at 30–1000 sperm per 100 µl of extract.

## Imaging

For imaging, 3 µl of extract was spread on the bottom of a well of a Corning 96-well polystyrene plate (#3368) and covered with 200 µl of heavy mineral oil (Sigma). The extract was imaged in time-lapse using a Leica epifluorescence microscope and a 5x objective. Acquisition frequency varied between 30 s to 2 min per frame.

## Image analysis

Images were automatically stitched, segmented, and analyzed using a custom-made pipeline scripted in Matlab. The code, as well as videos of all experiments analyzed and the original microscopy image stacks, are available from the Stanford Digital Repository (https://purl.stanford.edu/fm814ch0699). Mitotic sources and nuclei were located automatically and were checked manually. Centrosome positions were determined manually. Because the initiation time for the interphase of the first cycle was unknown, all of the analysis was performed starting from the first mitosis.

## Bootstrapping calculation

Bootstrapping was performed by randomizing the locations of mitotic initiation for a cycle, determining how many of these randomized sources were within 100 µm of a nucleus or a centrosome, and then repeating the procedure 100,000 times. The p-value for that cycle was calculated as the number of iterations in which there were as many or more sources in proximity by random, divided by 100,000.

## Acknowledgements

We thank Yuping Chen and Shixuan Liu for comments on the manuscript, and the Ferrell and Stearns labs for helpful discussions. This work was supported by grants from the National Institutes of Health (T32 GM007276-43, R01 GM110564, R35 GM131792, and R35 GM120286).

## Additional information

### Funding

| Funder | Grant reference number | Author |
|---|---|---|
| National Institutes of Health | R01 GM110564 | James E Ferrell |
| National Institutes of Health | R35 GM131792 | James E Ferrell |
| National Institutes of Health | R35 GM120286 | Tim Stearns |
| National Institutes of Health | GM007276 | Garrison K Buss |

The funders had no role in study design, data collection and interpretation, or the decision to submit the work for publication.

### Author contributions

Oshri Afanzar, Conceptualization, Software, Formal analysis, Investigation, Visualization, Writing - original draft, Writing - review and editing; Garrison K Buss, Investigation, Writing - review and editing; Tim Stearns, Supervision, Funding acquisition, Writing - review and editing; James E Ferrell Jr,

Conceptualization, Supervision, Funding acquisition, Visualization, Writing - original draft, Writing - review and editing

### Author ORCIDs
Oshri Afanzar (iD) https://orcid.org/0000-0001-9690-3180
Tim Stearns (iD) https://orcid.org/0000-0002-0671-6582
James E Ferrell Jr (iD) https://orcid.org/0000-0003-4767-3926

### Ethics
Animal experimentation: This study was performed in strict accordance with the recommendations in the Guide for the Care and Use of Laboratory Animals of the National Institutes of Health. All of the animals were handled according to approved institutional animal care and use committee (IACUC) protocols Stanford University (assurance no. A3213-01, protocol 13307).

### Decision letter and Author response
Decision letter https://doi.org/10.7554/eLife.59989.sa1
Author response https://doi.org/10.7554/eLife.59989.sa2

## Additional files

### Supplementary files
• Transparent reporting form

### Data availability
All data and code used in the analysis are available from the Stanford Digital Repository (https://purl.stanford.edu/fm814ch0699) for purposes of reproducing or extending the analysis. All materials will be made available by the authors upon request.

The following dataset was generated:

| Author(s) | Year | Dataset title | Dataset URL | Database and Identifier |
|---|---|---|---|---|
| Afanzar O, Buss GK, Stearns T, Ferrell JE | 2020 | The nucleus serves as the pacemaker for the cell cycle | https://purl.stanford.edu/fm814ch0699 | Stanford Digital Repository, fm814ch0699 |

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
