## [Decision Letter]

**Acceptance summary:**

Your paper presents beautiful experiments on the temporal organization of mitosis in a spatially extended extract system, thus providing important novel insights on where mitosis is initiated. The conclusion that mitosis start in or around nuclei in your system is very interesting. The combination of imaging and quantitative analysis is very exciting.

**Decision letter after peer review:**

Thank you for submitting your article "The nucleus serves as the pacemaker for the cell cycle" for consideration by *eLife*. Your article has been reviewed by three peer reviewers, including Stefano Di Talia as the Reviewing Editor and Reviewer #1, and the evaluation has been overseen by Anna Akhmanova as the Senior Editor. The following individual involved in review of your submission has agreed to reveal their identity: William M Bement (Reviewer #3).

The reviewers have discussed the reviews with one another and the Reviewing Editor has drafted this decision to help you prepare a revised submission.

Summary:

This is a very interesting and well-executed study on the timing of mitosis in spatially-extended system. The paper reaches the conclusion that in large cells the nucleus serves as the pacemaker of mitosis. That is a novel and important insight.

Essential revisions:

We would like you to follow reviewer 1 suggested edits to the paper with the goal of putting your new data in the context of previous literature.

Both reviewer 1 and 3 suggest that you add a longer and better Discussion. The field of mitotic waves and synchronization is becoming mature and it seems appropriate to discuss your new results in the broader context and drawing parallel and differences with other model systems.

Reviewer 2 also has few suggestions for improving the paper.

Finally, you will see that reviewer 3 asks for an experiment blocking the centrosome cycle with a drug. We all agree that such experiment will significantly improve the paper but it is not strictly required for acceptance. If you decided not to perform the experiment, it would be nice to at least cite the suggested experiment in the Discussion as a possible follow-up experiment.

Reviewer #1:

The manuscript by Afanzar et al., addresses the mechanisms of spatial control of mitosis. The authors build on previous findings from their group showing the existence of mitotic waves in *Xenopus* extracts and the ability of extracts to organize their cytoplasm in structures resembling those observed in syncytia (see energids in fly embryos). By analyzing the timing and pattern of mitosis in two-dimensional extracts the authors conclude that mitosis starts at the nucleus. This is an interesting finding and the experiments described in the paper are creative and well-executed. However, there are two important points (points 1 and 2) where the presentation of this paper needs to be improved as well as few more minor points where some discussion might be interesting.

1) The authors contrast their results to previous findings from the Pines' group which found no difference in the activity of Cdk1 in the nucleus and the cytoplasm. However, there is a likely possibility that the difference in the findings of this paper to the previous study by Pines' group can be explained by the largely different spatial dimensions of the two systems. It is possible that nuclear and cytoplasmic dynamics are correlated over small spatial scales but not large ones. This should be acknowledged. The importance of (individual) nuclei might become predominant in systems where inter-nuclear distances are very large.

2) The authors suggest that in the early stages of their extracts, waves are fast with several origins. They use this observation to argue that those waves might be phase waves. A mechanism for the generation of phase waves by gradients of Cdk1 activity has been described by Vergassola et al., 2018. This paper should be cited, and the possible explanation acknowledged. Notice that the recent paper by Nolet et al., also likely observed a similar transition from phase (sweep) waves to trigger waves.

3) Figure 1F shows that the nuclei in the extract are unable to self-organize in a regular pattern and generate regions of very different nuclear densities. It would be interesting to point this out and contrasts to systems like the fly embryo where nuclei self-organize in equally-spaced pattern. Are the nuclei just amplifying inhomogeneity in the initial conditions?

4) In Figure 2D the authors compared the probability of observing a source at a given distance from a nucleus and see that sources are enriched near nuclei. This is convincing. However, I wonder whether a model in which the sources are randomly distributed is too conservative. Ultimately, when nuclei are very sparse it might take forever for them to spread a wave to a region that is far away. Probably an analysis that keeps that into account would give stronger statistics.

5) It would be nice if the statement that the wave of mitotic exit travels slower than the one of mitotic entry was quantified in some statistical way. What are the error bars on the wave speeds?

Reviewer #2:

The paper studies the initiation of mitotic (trigger) waves through in vitro experiments. The main conclusions overlap with the recent paper by Nolet et al.,: the nucleus serves as a pacemaker that triggers waves and speeds up the cell cycle. Nevertheless, I find the Results of the current paper more direct and compelling. Nolet et al., work in an effectively 1D system, in contrast to the 2D system here. The data analysis of the current paper is fully automated, in contrast to that of Nolet et al., which was hard to parse. Most importantly, here the data analysis involves little modeling, relying instead on basic statistical analysis. In contrast, Nolet et al., have to resort to simulations of rather complex mathematical models in order to validate their conclusions.

The authors show that nuclei are significantly more likely to be wave initiation centers, and that this effect becomes more pronounced in later cycles. They also perform beautiful experiments in which they decouple the relative contributions of centrosomes from nuclei (again, well beyond what was achieved in the Nolet et al., paper), thereby showing that nuclei help initiate waves and speed up the cell cycle, while the centrosomes play a role in forming compartments. Overall, the paper is clearly written and accessible, and the results are of broad interest. For these reasons, I recommend publication in *eLife*.

Reviewer #3:

Entry into and exit from mitosis is driven by activation and inactivation of Cyclin dependent kinase 1 (Cdk1), respectively. It was previously demonstrated that Cdk1 activation in cell free frog egg extracts occurs as a bistable "trigger wave" and proposed, based on analysis of surface contraction waves in intact *Xenopus* eggs that the nucleus acts as a pacemaker for such waves (Chang and Ferrell,, 2013). A very recent study (Nolet et al., 2020) tested this hypothesis and concluded that, yes, nuclei do indeed serve as pacemakers for Cdk1 waves in *Xenopus* egg extracts and thus, presumably, intact cells as well.

In the current study, Afanzar et al., confirm the results of the recent e*Life* paper using the same basic approach (ie imaging Cdk1 waves in frog egg extracts containing nuclei and a probe for nuclear assembly) but with several differences in the assays and the manipulations. The assay differences of most interest are that Afanzar use a fluorescent microtubule binding probe for all of their experiments and use a quasi-2D environment (the bottom of wells in 96-well plates) whereas Nolet et al., (the authors of the recent paper) used nuclei themselves and quasi-1D environments (tubes of varying widths). The use of fluorescent tubulin and the 2D environment makes visualization of the waves quite straightforward and, because microtubules will cycle in the absence of nuclei or centrosomes, doing experiments with and without added nuclei or centrosomes is also straightforward. The manipulation differences of most interest are that Afanzar et al., included experiments with purified centrosomes and purified centrosomes mixed with phage DNA which, in principle, made it easier to rule out the centrosome as the pacemaker.

Afanzar et al., make a convincing case that nuclei do indeed serve as pacemakers in the extract system, thus confirming the results of Nolet et al. Afanzar et al., also clearly demonstrate that centrosomes alone do not serve as pacemakers, a point directly addressed by Nolet et al., and one other feature stands out: Afanzar et al., provide a welcome description of the rich variability of the system, which is made possible by the microtubule probe and the 2D environment. Thus, we learn that phase waves can coexist (in the same extract) with trigger waves, we learn that the trigger waves can arise at the edge of the well, and we learn that in the complete absence of added nuclei, mitotic sources can develop spontaneously and persist for several cell cycles.

I have one experimental suggestion. The results with the phage DNA-generated nuclei are puzzling in that while these can clearly serve as mitotic sources, they just as clearly have a far more modest effect in terms of cell cycle acceleration than the demembranated sperm. Consequently, it could be argued that while isolated centrosomes alone cannot serve as mitotic sources, a centrosome closely opposed to a nucleus makes it a better source. Unfortunately, the experiment where the phage DNA and the purified centrosomes are combined doesn't really help, in that there is no evidence that the two are close to each other once added to the extract, as they would be in a real cell or, for that matter, an extract seeded with demembranated sperm.

If, however, the authors were able to block centrosome replication by adding a pharmacological inhibitor of centriole replication (e.g. centrinone), and if centrosomes are completely superfluous, it would be expected that the mitotic sources that develop after several cell cycles in sperm-supplemented extras would be just as enthusiastic as those formed in the absence of the replication inhibitor.

[Editors' note: further revisions were suggested prior to acceptance, as described below.]

Thank you for submitting your article "The nucleus serves as the pacemaker for the cell cycle" for consideration by *eLife*. Your article has been reviewed by three peer reviewers, including Stefano Di Talia as the Reviewing Editor and Reviewer #1, and the evaluation has been overseen by Anna Akhmanova as the Senior Editor. The following individuals involved in review of your submission have agreed to reveal their identity: Ariel Amir (Reviewer #2); William M Bement (Reviewer #3).

The reviewers have discussed the reviews with one another and the Reviewing Editor has drafted this decision to help you prepare a revised submission.

Summary:

This revised manuscript is significantly improved. There is only one point that needs to be addressed before publication.

Essential revisions:

The results of Vergassola et al. are improperly cited. These authors did not observe gradients of cyclins but rather gradients of Cdk1 activity that, by having the Cdk1 levels raising uniformly over time, produce wave-like spreading of mitotic events. This point should be revised. Also, the theory by Vergassola et al., applies to any transient bistable system (including models of *Xenopus*' cell cycle) and predicts a switch from phase to trigger waves as cycles slow down. Readers will benefit from having that pointed out clearly.

Reviewer #1:

This revised manuscript is significantly improved, and the comments made in the previous version have been addressed. There is one place where the authors need to be more precise about their language.

Results of Vergassola et al., are improperly cited. These authors did not observe gradients of cyclins but rather gradients of Cdk1 activity that, by having the Cdk1 levels raising uniformly over time, produce wave-like spreading of mitotic events. This point should be revised. Also, the theory by Vergassola et al., applies to any transient bistable system (including models of *Xenopus*' cell cycle) and predicts a switch from phase to trigger waves as cycles slow down. Readers will benefit from having that pointed out clearly.

Reviewer #2:

The authors have addressed my previous comments and concerns.

Reviewer #3:

The authors have addressed all of my concerns.

---

## [Author Response]

Essential revisions:We would like you to follow reviewer 1 suggested edits to the paper with the goal of putting your new data in the context of previous literature.

Done (see below).

Both reviewer 1 and 3 suggest that you add a longer and better Discussion. The field of mitotic waves and synchronization is becoming mature and it seems appropriate to discuss your new results in the broader context and drawing parallel and differences with other model systems.

Done (see below).

Reviewer 2 also has few suggestions for improving the paper.

Done (see below).

Finally, you will see that reviewer 3 asks for an experiment blocking the centrosome cycle with a drug. We all agree that such experiment will significantly improve the paper but it is not strictly required for acceptance. If you decided not to perform the experiment, it would be nice to at least cite the suggested experiment in the Discussion as a possible follow-up experiment.

We have carried out preliminary experiments, and as far as they go they do support the hypothesis that the centrosomes contribute little if anything to the function of nuclei as trigger wave sources, but there are aspects of the experimental results that we do not understand completely and would like to explore further before publishing.

All of these changes are discussed in more detail below.

Reviewer #1:The manuscript by Afanzar et al., addresses the mechanisms of spatial control of mitosis. The authors build on previous findings from their group showing the existence of mitotic waves in *Xenopus* extracts and the ability of extracts to organize their cytoplasm in structures resembling those observed in syncytia (see energids in fly embryos). By analyzing the timing and pattern of mitosis in two-dimensional extracts the authors conclude that mitosis starts at the nucleus. This is an interesting finding and the experiments described in the paper are creative and well-executed. However, there are two important points (points 1 and 2) where the presentation of this paper needs to be improved as well as few more minor points where some discussion might be interesting.1) The authors contrast their results to previous findings from the Pines' group which found no difference in the activity of Cdk1 in the nucleus and the cytoplasm. However, there is a likely possibility that the difference in the findings of this paper to the previous study by Pines' group can be explained by the largely different spatial dimensions of the two systems. It is possible that nuclear and cytoplasmic dynamics are correlated over small spatial scales but not large ones. This should be acknowledged. The importance of (individual) nuclei might become predominant in systems where inter-nuclear distances are very large.

We agree and have revised the wording (Introduction) to acknowledge this.

2) The authors suggest that in the early stages of their extracts, waves are fast with several origins. They use this observation to argue that those waves might be phase waves. A mechanism for the generation of phase waves by gradients of Cdk1 activity has been described by Vergassola et al., 2018. This paper should be cited, and the possible explanation acknowledged. Notice that the recent paper by Nolet et al., also likely observed a similar transition from phase (sweep) waves to trigger waves.

We agree, and in fact even before the Nolet paper was published, we had speculated that in the earliest cell cycles in extracts in Teflon tubes, the mitotic waves might be phase waves (Chang and Ferrell, 2013). As suggested, we now cite the two Vergassola/Di Talia et al., papers on *Drosphila* Cdk1 waves in the Results and in the Discussion.

3) Figure 1F shows that the nuclei in the extract are unable to self-organize in a regular pattern and generate regions of very different nuclear densities. It would be interesting to point this out and contrasts to systems like the fly embryo where nuclei self-organize in equally-spaced pattern. Are the nuclei just amplifying inhomogeneity in the initial conditions?

Right when we put nuclei into an extract at low concentrations. On the other hand, if you put nuclei in an extract at high enough concentrations, they do end up migrating on microtubules and forming a fairly regularly spaced array of cell-like compartments with nuclei, sometimes with multiple nuclei per cell-like compartment. This is discussed in detail in Cheng and Ferrell, 2019. We think that this regime may be more like what is happening in the fly embryo.

Likewise, in the experiments we present here, as the nuclei divide the daughter nuclei form a regularly spaced area of cell like compartments much as they do in the fly embryo (e.g. what was Figure 1F and is now Figure 1C). So, the nuclei do not spread out evenly through the whole area of the well, but in the part of the well where there are nuclei, they do end up positioned in a regular way.

The nuclei do amplify inhomogeneity; the mitotic wave sources get “deeper”, in the topographic map sense, as the cycles proceed. However, it is not clear they amplify initial inhomogeneity. There is not always an obvious connection between where mitosis initiates in the first cycle, before nuclei have formed, and the positions of the nuclear sources in the subsequent cycles. Take Figure 2C for example. In the first mitosis, there are numerous shallow sources, some at the boundary (red) and some not associated with an obvious structure (blue). Then by mitosis 3, there are nuclei present, and the “deepest” sources are associated with nuclei. But the nuclear are not generally positioned where initial blue sources were. So, the pair of nuclear sources near the edge at 11:00, the pair at 2:00, the single deep source at 4:00, and the single deep source at 7:00, all reside in regions where there was no source in mitosis 1.

We now point this out in subsection “Mitotic sources are often associated with nuclei and/or centrosomes”.

4) In Figure 2D the authors compared the probability of observing a source at a given distance from a nucleus and see that sources are enriched near nuclei. This is convincing. However, I wonder whether a model in which the sources are randomly distributed is too conservative. Ultimately, when nuclei are very sparse it might take forever for them to spread a wave to a region that is far away. Probably an analysis that keeps that into account would give stronger statistics.

The bootstrapping results seem pretty easy to understand. We are not sure how to implement the non-random source model the reviewer suggests. If you think this would be helpful for the revised paper, could you explain further what you are thinking here?

5) It would be nice if the statement that the wave of mitotic exit travels slower than the one of mitotic entry was quantified in some statistical way. What are the error bars on the wave speeds?

As suggested, we now include averages and standard deviations for the mitotic entry wave speed and the aster growth speed (subsection “Spatial dynamics of mitotic initiation”). We also added a new figure panel (Figure 1F) that shows averaged entry and exit waves from 33 mitoses.

Reviewer #3:Entry into and exit from mitosis is driven by activation and inactivation of Cyclin dependent kinase 1 (Cdk1), respectively. It was previously demonstrated that Cdk1 activation in cell free frog egg extracts occurs as a bistable "trigger wave" and proposed, based on analysis of surface contraction waves in intact *Xenopus* eggs that the nucleus acts as a pacemaker for such waves (Chang and Ferrell, 2013). A very recent study (Nolet et al., 2020) tested this hypothesis and concluded that, yes, nuclei do indeed serve as pacemakers for Cdk1 waves in *Xenopus* egg extracts and thus, presumably, intact cells as well.In the current study, Afanzar et al., confirm the results of the recent eLife paper using the same basic approach (ie imaging Cdk1 waves in frog egg extracts containing nuclei and a probe for nuclear assembly) but with several differences in the assays and the manipulations. The assay differences of most interest are that Afanzar use a fluorescent microtubule binding probe for all of their experiments and use a quasi-2D environment (the bottom of wells in 96-well plates) whereas Nolet et al., (the authors of the recent paper) used nuclei themselves and quasi-1D environments (tubes of varying widths). The use of fluorescent tubulin and the 2D environment makes visualization of the waves quite straightforward and, because microtubules will cycle in the absence of nuclei or centrosomes, doing experiments with and without added nuclei or centrosomes is also straightforward. The manipulation differences of most interest are that Afanzar et al., included experiments with purified centrosomes and purified centrosomes mixed with phage DNA which, in principle, made it easier to rule out the centrosome as the pacemaker.Afanzar et al., make a convincing case that nuclei do indeed serve as pacemakers in the extract system, thus confirming the results of Nolet et al. Afanzar et al., also clearly demonstrate that centrosomes alone do not serve as pacemakers, a point directly addressed by Nolet et al., and one other feature stands out: Afanzar et al., provide a welcome description of the rich variability of the system, which is made possible by the microtubule probe and the 2D environment. Thus, we learn that phase waves can coexist (in the same extract) with trigger waves, we learn that the trigger waves can arise at the edge of the well, and we learn that in the complete absence of added nuclei, mitotic sources can develop spontaneously and persist for several cell cycles.I have one experimental suggestion. The results with the phage DNA-generated nuclei are puzzling in that while these can clearly serve as mitotic sources, they just as clearly have a far more modest effect in terms of cell cycle acceleration than the demembranated sperm. Consequently, it could be argued that while isolated centrosomes alone cannot serve as mitotic sources, a centrosome closely opposed to a nucleus makes it a better source. Unfortunately, the experiment where the phage DNA and the purified centrosomes are combined doesn't really help, in that there is no evidence that the two are close to each other once added to the extract, as they would be in a real cell or, for that matter, an extract seeded with demembranated sperm.If, however, the authors were able to block centrosome replication by adding a pharmacological inhibitor of centriole replication (e.g. centrinone), and if centrosomes are completely superfluous, it would be expected that the mitotic sources that develop after several cell cycles in sperm-supplemented extras would be just as enthusiastic as those formed in the absence of the replication inhibitor.

We have carried out the suggested experiment. We found that even at high concentrations of centrinone, sperm were still as good at initiating trigger waves as they were in its absence. This is consistent with the idea that centrosomes do not really help initiate trigger waves (at least not in any way we can measure).

But we also found effects of centrinone on the organization of the extract that we do not completely understand. They occur at somewhat higher concentrations of the drug than those needed to block centriole duplication but not so high that we can comfortably attribute the effects to some target other than Plx4. We plan to carry out additional experiments with other ways of monitoring centrosome duplication and other ways of compromising Plx4 activity, but for now would prefer not to mention the centrinone results until we understand them better.

[Editors' note: further revisions were suggested prior to acceptance, as described below.]

Essential revisions:The results of Vergassola et al. are improperly cited. These authors did not observe gradients of cyclins but rather gradients of Cdk1 activity that, by having the Cdk1 levels raising uniformly over time, produce wave-like spreading of mitotic events. This point should be revised. Also, the theory by Vergassola et al., applies to any transient bistable system (including models of *Xenopus'* cell cycle) and predicts a switch from phase to trigger waves as cycles slow down. Readers will benefit from having that pointed out clearly.Reviewer #1:This revised manuscript is significantly improved, and the comments made in the previous version have been addressed. There is one place where the authors need to be more precise about their language.Results of Vergassola et al., are improperly cited. These authors did not observe gradients of cyclins but rather gradients of Cdk1 activity that, by having the Cdk1 levels raising uniformly over time, produce wave-like spreading of mitotic events. This point should be revised. Also, the theory by Vergassola et al., applies to any transient bistable system (including models of *Xenopus'* cell cycle) and predicts a switch from phase to trigger waves as cycles slow down. Readers will benefit from having that pointed out clearly.

There was one remaining criticism that we change the wording of how we describe the Vergassola / Di Talia sweep wave paper. We have done that in subsection “Spatial dynamics of mitotic initiation” and in the Discussion.